# Guiding Offline Reinforcement Learning Using Safety Expert

**Richa Verma**[1,3], **Kartik Bharadwaj**[2,3], **Harshad Khadilkar**[1], **Balaraman Ravindran**[2,3]

[1]TCS Research
[2]Robert Bosch Centre for Data Science and Artificial Intelligence
[3]Department of Computer Science and Engineering, Indian Institute of Technology Madras, Chennai, India
richa.verma4@tcs.com, cs20s020@smail.iitm.ac.in

## Abstract

Offline reinforcement learning is used to train policies in situations where it is expensive or infeasible to access the environment during training. An agent trained under such a scenario does not get corrective feedback once the learned policy starts diverging and may fall prey to the overestimation bias commonly seen in this setting. This increases the chances of the agent choosing unsafe/risky actions, especially in states with sparse to no representation in the training dataset. In this paper, we propose to leverage a safety expert to discourage the offline RL agent from choosing unsafe actions in under-represented states in the dataset. The proposed framework in this paper transfers the safety expert's knowledge in an offline setting for states with high uncertainty to prevent catastrophic failures from occurring in safety-critical domains. We use a simple but effective approach to quantify the state uncertainty based on how frequently they appear in a training dataset. In states with high uncertainty, the offline RL agent mimics the safety expert while maximizing the long-term reward. We modify TD3+BC, an existing offline RL algorithm, as a part of the proposed approach. We demonstrate empirically that our approach performs better than TD3+BC on some control tasks and comparably on others across two sets of benchmark datasets while reducing the chance of taking unsafe actions in sparse regions of the state space.

## 1 Introduction

Reinforcement Learning (RL) has seen advancement and achieved great success in solving complex tasks with high dimensional state and action spaces, including games [Mnih et al., 2013, 2015, Lillicrap et al., 2015, Gu et al., 2017], and some tasks from robotics [Levine et al., 2016]. An RL agent trained in an online setting takes an action $a$ in state $s$ and interacts with the environment to observe a reward $r$. It then updates its policy based on the observed reward. However, it may be risky or costly to interact with the environment repeatedly in real-world situations. It may be infeasible in the cases where a high quality simulator is not available or cannot be built.

In offline RL (also known as batch RL), the agent is not allowed to interact with the environment. It has access to a fixed-sized dataset collected by any arbitrary policy which may or may not be known [Lange et al., 2012]. Real-world applications can benefit from this setting because access to the environment may be limited, challenging or not possible. Such applications which are already deployed can also generate datasets to learn from. Offline RL enables the use of such logged datasets for learning and can even allow us to leverage an expert in the form of a human operator, rule-based systems or a policy trained with a similar objective. Some approaches such as [Lee et al., 2020] show

Offline Reinforcement Learning Workshop at Neural Information Processing Systems, 2022.

that dataset collected by an expert during learning in an online setting can also be used, however, using the expert itself to facilitate learning in offline RL eliminates the need for data collection and is helpful in settings where data privacy needs to be enforced.

Overestimation of the values of out-of-distribution actions is a fundamental challenge in offline RL. This also applies to certain actions which can be deemed as "unsafe" in safety-critical applications such as autonomous driving, robotic learning, healthcare, etc. For robotic learning, the conditions for a safety breach during an episode are easier to define (e.g. recording how many times the robot has fallen, or a grasped object has been dropped). The challenge in this domain is to learn an optimal policy for a task while minimizing the frequency of above-mentioned instances of catastrophic failures during training.

In this paper, we study how to utilize a safety expert in an offline RL setting for states with high uncertainty to minimize failures during training. This safety expert isn't necessarily optimal and can be learned or defined by a rule-based system for each task without reference to the underlying task reward. We use a simple but effective approach to quantify the uncertainty of the states based on how frequent the visited states are in a given training dataset. This information is used to conservatively modify the critic target, therefore propagating it to the value function estimation. We believe that incorporating a safety expert in the form of a pre-trained teacher policy along with quantifying state uncertainty can be effective in this setting. It reduces the chances of the offline RL agent engaging in potentially risky exploratory behavior, thus enabling robotic learning from massive datasets. We show that it can allow the agent to learn safe behavior without explicitly defining constraints on actions, which can be hard to do in an offline setting.

Our goal is to selectively utilize a safe teacher policy to reduce the chances of risky/unsafe behavior encountered during the deployment of a learned offline RL policy while still maintaining high performance. Our main contributions are summarized below:

- We propose a framework called **Guided TD3+BC** that trains an agent to learn efficiently from an offline dataset while leveraging a safety expert in regions of high uncertainty.
- We evaluate our approach on a set of datasets from the D4RL benchmark of continuous control tasks [Fu et al., 2020] and show that the proposed framework either performs better or comparably to TD3+BC [Fujimoto and Gu, 2021], a popular SOTA offline RL algorithm on most of the tasks.

## 2 Related Work

**Offline RL**. The existing offline RL methods mainly use some approach that allows the learned policy to stay close to the data collection policy. There are various ways of implementing this. One way is to estimate the behavior policy and then learning a parameterized policy [Fujimoto et al., 2019b, Ghasemipour et al., 2020]. Another line of works uses divergence regularization [Jaques et al., 2019, Kumar et al., 2019, Fujimoto et al., 2019a] to keep the two policies close to each other. Some other works suggest the use of a weighted version of behavior cloning to encourage choosing actions with high advantage [Peng et al., 2019, Nair et al., 2020] or use uncertainty as weight of a state-action pair before making updates[Wu et al., 2021]. Some methods incorporate the notion of safety and modify the set actions that can be chosen based on their counts [Laroche et al., 2019]. promising direction of literature looks at using pessimism and implementing divergence regularization as a part of value estimation [Kumar et al., 2020, Buckman et al., 2020]. The goal of this work is different from these works which focus on developing RL alorithms specifically for an offline setting. We study knowledge transfer from a safety expert to an agent learning in the offline setting.

**Reinforcement Learning from Demonstration.** RL literature has many examples of learning from teacher policies or demonstrations in an online setting, especially in hard exploration environments. There are policy distillation techniques [Parisotto et al., 2015, Rusu et al., 2015] for training student networks such that their outputs (e.g., Q-values) are similar to those of teacher networks. Learning from demonstrations is another promising area. A replay buffer in an off-policy RL setting can be

used to hold teacher demonstrations, which can be combined with samples generated by a student agent during training. DQfD [Hester et al., 2018] and Ape-X DQfD [Pohlen et al., 2018] are some of the examples of such methods for a discrete setting while methods suggested by [Nair et al., 2018, Vecerik et al., 2017] work for continuous control tasks.

## 3 Proposed Approach

In offline RL, the problem of extrapolation error [Fujimoto et al., 2019b] is prevalent which means that the agent is unable to evaluate out-of-distribution actions properly. Our focus is on designing a framework to discourage the agent from selecting unsafe OOD actions while trying to learn an optimal policy from the dataset. We present such a framework that requires minimal modifications to a pre-existing offline RL algorithm. Our framework builds on top of TD3+BC [Fujimoto and Gu, 2021]. We modify the critic target term to include state uncertainty. We also include a regularization term to push the offline policy towards the safety expert in states with poor confidence. The safety expert can be defined by any rule-based system or a pre-trained policy. We denote the agent's confidence w.r.t a state as $conf(s) \in [0, 1]$, where the confidence is computed by using SimHash algorithm [Tang et al., 2017]. SimHash uses Locality-Sensitive Hashing (LSH) to convert continuous, high-dimensional data to discrete hash codes. LSH preserves the distances among data points, such that those with similar hashes are close to each other. We use SimHash which is a computationally efficient LSH technique and it measures the similarity of the states contained in the training dataset $\mathcal{D}$ by angular distance. Here, we can use any technique which can transform the high-dimensional continuous state space into discrete bins based on their closeness. The following equation shows how hash codes are computed:

$$\mu(s) = sgn(Ag(s)) \in [-1, 1]^k. \tag{1}$$

where $A \in \mathcal{R}^{k \times d}$ is a matrix with each entry drawn i.i.d. from a standard Gaussian and $g : S \to \mathcal{R}^{\mathcal{D}}$ is a preprocessing function. The dimension of binary codes is $k$ and it controls the granularity of the state space discretization. This algorithm was originally used as an exploration method but we use it to bin the states contained in the dataset $\mathcal{D}$ into hash codes of size $k$. We use $k = 50$ for all the tasks after careful experimentation with multiple tasks. We populate the hashtable by recording the counts of states mapped to each hash code, before training an agent. We normalize the state count values by using max-min normalization. Further, we query the hashtable to retrieve these counts during training and use the values as $conf(s)$ in the below critic target update equation:

$$Q(s, a) = r + \gamma * \max_{a'} Q(s', a') \underbrace{- (1 - conf(s)) * (a - \pi_T(s))^2}_{\text{uncertainty weighted learning from the safety expert}}. \tag{2}$$

where $\pi_T(s)$ is a teacher policy used as the safety expert. It is trained in an online setting using a continuous control algorithm known as TD3 [Fujimoto et al., 2018]. More details on training the policy $\pi_T(s)$ to be safe are provided in the next section.

Note that the value of $conf(s)$ is lower for under-represented states in the given dataset $\mathcal{D}$ and the lower the confidence, the higher will be the push towards the safety expert, $\pi_T(s)$. Also, the modified update equation reduces the values of all the $(s, a)$ pairs in the dataset except the ones with the action suggested by the safety expert. This discourages the agent from picking unsafe action values in regions of high uncertainty. This completes the description of our framework called **Guided Offline RL** which involves making a few small, but effective, modifications to TD3+BC.

## 4 Experiments

We evaluate our proposed approach on the D4RL benchmark of OpenAI gym MuJoCo tasks [Fu et al., 2020]. We use the TD3+BC algorithm trained on MuJoCo tasks (Hopper-v2 and Walker2d-v2) as the baseline. We train a teacher policy $\pi_T$ to be used as the safety expert using TD3 for 1M online steps. For the policy to be safe, we add a step penalty of the form $ctrl\_cost\_weight * sum(action^2)$ which is simply a cost for penalizing the agent if it takes actions that are too large. We observe that by doing so, we can discourage the agent from applying high values of torques to various joints of a MuJoCo robot and hence prevent it from making jittery moves. We choose $ctrl\_cost\_weight$ as 0.1 and 0.01 for Hopper-v2 and Walker2d-v2, respectively, after tuning. These environments have

in-built rewards which penalise the agent when it falls or when the height of the top (along the z-axis) becomes too high or too low. Further, we train the offline RL agent on various environment-dataset pairs using the safety expert policy $\pi_T$ as a part of the framework described in the previous section.

| Dataset | Environment | TD3+BC | Guided TD3+BC |
|---|---|---|---|
| Random | Hopper-v2 | 8.53 $\pm$0.23 | 6.03 $\pm$2.03 |
| | Walker2d-v2 | 0.95 $\pm$0.33 | 2.83 $\pm$3.57 |
| Medium | Hopper-v2 | 60.12 $\pm$1.35 | 57.77 $\pm$ 3.07 |
| | Walker2d-v2 | 86.17 $\pm$0.3 | 83.78 $\pm$2.91 |
| Medium-Replay | Hopper-v2 | 56.71 $\pm$19.16 | 85.61 $\pm$5.14 |
| | Walker2d-v2 | 73.56 $\pm$11.19 | 84.67 $\pm$0.77 |
| Medium-Expert | Hopper-v2 | 95.16 $\pm$9.85 | 106.11 $\pm$5.92 |
| | Walker2d-v2 | 110.26 $\pm$0.65 | 110.6 $\pm$0.21 |
| Expert | Hopper-v2 | 110.97 $\pm$ 1.45 | 111.62 $\pm$0.37 |
| | Walker2d-v2 | 110.12 $\pm$ 0.47 | 109.91 $\pm$0.13 |
| | Total | 712.55 $\pm$44.98 | 758.93 $\pm$24.12 |

Table 1: Average normalized score using the D4RL -v2 datasets. The highest performing scores are highlighted. $\pm$ captures the standard deviation over seeds. TD3+BC algorithm is re-run using author-provided implementation. The results are after averaging over the final 10 evaluations and 3 seeds. No additional hyperparameter tuning was performed. TD3+BC and Guided TD3+BC achieve comparable performance.

We use the author-provided implementations for both TD3 and TD3+BC. We use the same base hyperparameters as the respective authors for these algorithms and train the baseline and the offline RL agent for three random seeds. In all experiments, the offline agent and the baseline agent do 10 evaluation episodes after every 5000 offline training steps till they reach 1M training steps. We use the normalized score from D4RL for evaluation and we average the scores of all seeds for each environment. We report the final performance results in Table 1. In Figure 1, we report the percentage difference between Guided Offline RL and TD3+BC w.r.t. the total number of times the agent falls or its agent's height crosses the safe range (Walker2d-v2) during all the evaluation episodes occurring within 1M training steps. We also report the percentage difference between the cumulative sum of the actions across all evaluation steps for each dataset-environment pair.

Our results show that including a safe teacher policy can help in reducing the number of falls that an agent has. We also show that the approach can keep the sum of actions low in most cases, as compared to the baseline. The proposed approach works better in reducing the number of falls in Walker2d environment as compared to Hopper (left). Here, our approach works better for the dataset-environment pairs for which the dataset collection policy is less similar to the safe teacher policy. The reduction of the cumulative sum of the actions is more pronounced for Hopper. We believe that if $\pi_T$ is trained using a constrained method to keep the sum of the actions low, the results could be better. We find our approach only marginally increases the training time as compared to that of the baseline. All run time experiments were run with a single GeForce GTX 1080 Ti GPU and an Intel(R) Xeon(R) CPU E5-2640 v4.

## 5 Conclusion

In this paper, we present Guided Offline RL framework which relies on state uncertainty estimation and safety expert knowledge to discourage an offline RL agent from choosing risky/unsafe actions. We have shown that an existing offline RL algorithm called TD3+BC can be easily modified to design the proposed framework. Our experiments show that our approach performs comparably or better on multiple MuJoCo tasks from D4RL benchmark while trying to minimize unsafe incidents during evaluation. We believe that our framework can be used as an add-on to help to achieve better results

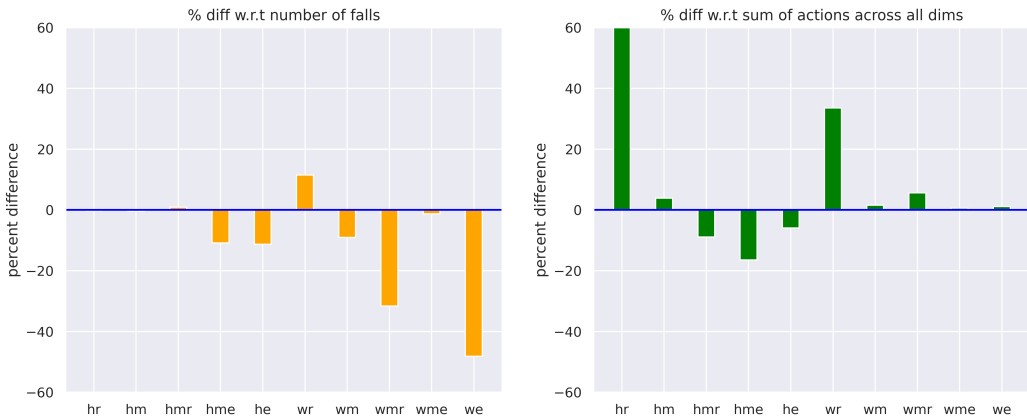

Figure 1: Percent difference of performance of Guided Offline RL w.r.t baseline TD3+BC algorithm. Here, h = Hopper-v2, w = Walker2d-v2, r = random, m = medium, mr = medium-replay, me = medium-expert, e = expert. The proposed approach works better in reducing the number of falls in Walker2d environment as compared to Hopper (left). The reduction of the cumulative sum of the actions is more pronounced for Hopper (right).

while adhering to safety. As future work, we consider using other forms of the safety expert such as human interventions, heuristics etc. and evaluate them on a diverse set of safety tasks. We also plan on studying the effectiveness of the framework when coupled with other SOTA offline RL algorithms.

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
