# OpenReview forum: "Guiding Offline Reinforcement Learning Using Safety Expert"
_NeurIPS.cc/2022/Workshop/Offline_RL — Offline RL Workshop NeurIPS 2022_

### Official Review · Reviewer_FpBP · 2022-10-17
**Better motivation and experimental evaluation needed**

**Rating:** 3
**Confidence:** 3

**Review:**

Summary: The paper proposes a method to learn safe policy from a offline dataset. Authors propose to guide the offline learning agent with a safe teacher policy at the states infrequently visited in the offline dataset. The method is validated on Hopper and Walker environments.


Strengths:
1.  Proposes a method to solve the important problem of learning safe policies from offline datasets.

Weaknesses:

1. My biggest concern is the fundamental assumption of the paper that states with low uncertainty are likely to be unsafe. The claim needs to be verified or demonstrated that it is relevant in practical applications. As a counterexample, an offline dataset can have all unsafe trajectories starting at a particular state s. Thus there is very low uncertainty in s but the current method does not do anything to prevent unsafe actions from s.
2. Line 95 states that the paper aims to present a method that discourages the agent from selecting OOD actions. This is already a part of the offline RL objective in methods like IQL, BRAC, CQL, etc. It might help to better motivate the problem.
3. I am not sure if Table 1 is a fair evaluation. Section 4 describes that the safety policy is trained with the task reward with a step penalty. Thus the safety policy has quite a lot of information about solving the task. This information will also get leaked into Guided TD3+BC and allow the agent for being more robust at low empirical probability states. So this gives Guided TD3+BC an unfair advantage in terms of return comparison. I think only the safety results make sense in the setting considered by the authors.
4. Method lacks any theoretical guarantees on safety.

---

### Official Review · Reviewer_7RrQ · 2022-10-19
**Interesting direction, but could use better experiments and ablations**

**Rating:** 6
**Confidence:** 4

**Review:**

This paper focuses on an interesting and important problem setup that would be of interest to the broader research community. In particular, this paper explores how an existing suboptimal safety controller can be leveraged to improve the performance of offline RL and prevent catastrophic failures in safety-critical domains. This setting is quite realistic and practical and could be important in brining learning based approaches for control to safety-critical domains like autonomous driving.

This paper's approach is to add an uncertainty weighted penalty to the Q function based on the actions deviation from the safety policy's action. The uncertainty is based on state-visitation frequency estimated with SimHash.

The major weakness of this work is that they do not evaluate on any actual safety-critical domains. It seems like the main selling point of this line of work is that it could improve performance in settings where there exists states where one major mistake could lead to a catastrophic failure, and simple safety controllers exist, but are otherwise suboptimal. However, the only experiments are on D4RL mujoco, where I don't think this is the case.

The authors should report the performance of running the safety expert online in the results to make the comparison to TD3+BC more clear and fair. Additionally, I would be interested in seeing some ablations on how different ways of incorporating the safety expert + state-uncertainty into the algorithm affects performance.

Pros:
Interesting problem setup
Reasonable approach to incorporate safety expert

Cons:
Poor experimental domain based on motivation
Lacks appropriate fair comparisons (mainly the safety expert by itself)